Production of recombinant human epidermal growth factor fused with HaloTag protein and characterisation of its biological functions

Bai Mengru 1
Liu Yezhuo 2 3
Liu Hongyin 1
Jia Yangyang 1
Tian Xiangqin 1
http://orcid.org/0000-0001-8602-9629 Sun Changye 1 3 sunchangye-boy@163.com
1 Henan Key Laboratory of Medical Tissue Regeneration, Xinxiang Medical University , Xinxiang, Henan , China
2 Favorsun Medical Group , Suzhou , China
3 Shanghai Hongdu New Material Science and Technology , Shanghai , China
Oliveira Sonia
Electronic publication date: 2024 Jul 16
Publication date: 2024
Volume: 12
Electronic Location ID: e17806
Received 2024 Mar 13; Accepted 2024 Jul 3
Copyright: © 2024 Bai et al.
Copyright year: 2024
Copyright holder: Bai et al.
License: This is an open access article distributed under the terms of the Creative Commons Attribution License, which permits unrestricted use, distribution, reproduction and adaptation in any medium and for any purpose provided that it is properly attributed. For attribution, the original author(s), title, publication source (PeerJ) and either DOI or URL of the article must be cited.
License URL: https://creativecommons.org/licenses/by/4.0/

Keywords: Epidermal growth factor, HaloTag, Protein expression and purification, Cell viability, HaCaT, RNA sequencing

Funding: Xinxiang Medical University 300-505232 Science and Technology Commission of Shanghai Municipality 210H1029300 Henan Province Science-Technology Department 242102310413 This work was supported by Xinxiang Medical University (300-505232), the Science and Technology Commission of Shanghai Municipality (210H1029300) and the Henan Province Science-Technology Department (242102310413). The funders had no role in study design, data collection and analysis, decision to publish, or preparation of the manuscript.

==============================
Epidermal growth factor (EGF) protein is a crucial biomolecule involved in regulating cell growth, proliferation, migration and differentiation, which is used in various therapeutic applications, such as wound healing and tissue regeneration. The production of recombinant EGF is essential for studying its biological function and for its clinical translation. However, EGF protein expressed in prokaryotic cells often occurs in inclusion bodies, and co-expression with soluble tag protein is an effective method to prepare recombinant EGF. In this study, we expressed recombinant human EGF (rhEGF) fused to a HaloTag (Halo-rhEGF) and a large portion of Halo-rhEGF was found in the soluble fraction. Cell growth assay showed that the purified Halo-rhEGF protein could promote the proliferation of fibroblasts (NIH 3T3) and epithelial cells (HaCaT), and significantly increased their viability. Phosphorylation of the intracellular signaling proteins, ERK1/2 and c-Jun, was stimulated by treatment with Halo-rhEGF and the expression levels of proteins regulating cell proliferation were significantly increased. RNA sequencing analysis revealed that rhEGF could increase the transcription of genes enriched in ribosome generation and cell proliferation. Moreover, Halo-rhEGF can be labelled by HaloTag ligand for fluorescence imaging and can be slowly released in tissue repair by binding to anion biomaterials. In conclusion, HaloTag is an efficient fusion tag for rhEGF protein expression, purification and controlled release, and Halo-rhEGF can promote the proliferation and viability of epithelial and fibroblast cells.

Introduction

Human epidermal growth factor (EGF) is a cell growth factor that stimulates the growth and division of diverse mammalian cells (Cohen & Carpenter, 1975; Fu et al., 2018; Shin et al., 2023). EGF receptors belong to the tyrosine kinase receptor family, and binding to EGF can stimulate the target cells activation, which in return promotes tissue growth and regeneration (Kato et al., 1998; Yarden & Sliwkowski, 2001; Shin et al., 2023). Additionally, EGF also serves as a signaling molecule in various diseases and physiological processes, such as cancer and heart development (Yarden & Sliwkowski, 2001; Iwamoto & Mekada, 2006). Previous studies have demonstrated that recombinant EGF protein has great potential in clinical treatment of tissue injury, such as skin wound healing (Shin et al., 2023).

The precursor of human EGF comprises 1,207 amino acids containing multiple EGF-like peptides and a 6 kDa EGF polypeptide (Asn971-Arg1023) which has been widely used in the treatment of tissue repair (Cohen & Carpenter, 1975; Scott et al., 1985; Valcarce, Björk & Stenflo, 1999; Hori et al., 2007; Ferreira et al., 2022). Prokaryotic protein expression systems can reach high yield and be cost-effective for recombinant protein production (Chen, 2012). Recombinant EGF protein was expressed in Escherichia coli (E. coli), but the directly expressed protein often occurs in inclusion bodies (Ferrer Soler et al., 2003). There are multiple strategies to co-express EGF with soluble proteins, such as thioredoxin, SUMO, maltose-binding protein, Fh8 and glutathione S-transferase (Kim et al., 2021; Ferreira et al., 2022). These tag proteins could improve the solubility of recombinant EGF expressed in E. coli and could be easily removed by proteolytic cleavage, but a size exclusion chromatography is required for further purification of recombinant EGF (Kim et al., 2021; Ferreira et al., 2022).

HaloTag is a modified form of a bacterial enzyme known as haloalkane dehalogenase. It reacts with chloroalkane ligands to create a covalent bond, which is a representative of the enzyme’s normal catalytic cycle (Los et al., 2008). Researchers have utilized fluorescent dyes (Los et al., 2008) and quantum dots (Zhang et al., 2006) carrying a chloroalkane group to label HaloTag fusion proteins for fluorescence imaging. This method is highly adaptable as it combines the advantages of a genetically encoded tag (HaloTag protein) with covalent binding to its fluorescent ligand. In the previous study, we had shown that HaloTag also significantly increased the solubility and yield of fibroblast growth factor (FGF) proteins such as FGF8, FGF17, and FGF18 (Sun et al., 2015). Moreover, there is a certain amount of negatively charged residues on the surface of HaloTag protein, which makes it easy to be purified by ion exchange chromatography (Sun et al., 2015).

In this study, we applied this N-terminal HaloTag vector to co-express HaloTag and recombinant human EGF (Halo-rhEGF). It was found that a large amount of Halo-rhEGF protein was expressed as a soluble form and the purified Halo-rhEGF exhibited the same biological functions as commercial rhEGF. Further study showed that Halo-rhEGF could stimulate the proliferation of NIH 3T3 and HaCaT and protect HaCaT from photodamage caused by ultraviolet B (UVB) irradiation. In addition, HaloTag protein can bind to cation-rich biomaterials to maintain the slow release of EGF. Thus, HaloTag is more than just a tag protein for fluorescence probe labeling, which can also be used as a soluble tag for protein expression and as a binding tag for sustained release of protein drugs.

Materials and Methods

Materials

The enzymes (FastDigest NcoI: #FD0574 and FastDigest BamHI: #FD0054) used for DNA digestion were purchased from Thermo Fisher Scientific (Waltham, MA, USA). Cloning and Assembly Kit for DNA recombination (CU101-01) and E. coli strains (Trans10: CD101-01 and BL21 (DE3): CD601-02) were bought from Transgen Biotech (Beijing, China). Nickel Sepharose and Q Sepharose Fast Flow (code numbers: 17-5318-01 and 17-1014-01, GE Healthcare Life Sciences, Uppsala, Sweden) were used for recombinant Halo-rhEGF protein purification. The following materials were used for cell culture: Dulbecco’s modified Eagle’s medium (DMEM) containing 4.5 g/L glucose and 4 mM L-Glutamine (C11995500BT, Thermo Fisher Scientific, Waltham, MA, USA), fetal bovine serum (FBS, A6903FBS-500, Invigentech, Irvine, USA), trypsin and penicillin-streptomycin (catalog numbers: T1300 and P1400, Solarbio, Beijing, China), cell culture dishes and plates (corning, Shanghai, China). Anti-CCND1 (ET1601-31), Anti-ERK1/2 (ET1601-29) and Anti-c-Jun (ET1608-3) were purchased from HuaBio (Hangzhou, China). Anti-phos-ERK1/2 (#4377) and Anti-phos-c-Jun (#91952) were purchased from Cell Signaling Technology (Shanghai, China). Anti-HMGA1 (CY7237), Anti-MCM2 (CY5200) and Anti-PCNA (AB0051) were purchased from Abways (Shanghai, China). Integrin alpha-1 antibody (29042-1-AP) was purchased from Proteintech (Wuhan, China). U0126 (M1977) and SP6125 (M2076) were ordered from Abmole Bioscience (Shanghai, China). TEV protease (P2307), Hoechst 33342 (C1022), Calcein AM (C2012-0.1 ml) and ROS assay kit (S0033S) were ordered from Beyotime (Shanghai, China).

Construction of Halo-rhEGF expression vector

The peptide sequence of human EGF (NCBI Reference Sequence: NP_001954.2, Asn971-Arg1023) containing a C-terminal TEV cleavage site and a hexahistidine tag was uploaded to GeneOptimizer (Invitrogen GeneArt, Thermo Fisher, Shanghai, China) to generate a modified DNA sequence for EGF protein expression in E. coli. The synthesized DNA sequence was validated by DNA sequencing and cloned into the pET14b-Halo expression vector as described (Sun et al., 2015). Finally, the expression vector (pET-14b-Halo-rhEGF) was transformed into BL21 (DE3) for Halo-rhEGF expression.

Expression and purification of Halo-rhEGF

The bacteria containing Halo-rhEGF expression vector was cultured at 37 °C until the OD600 was between 0.4 and 0.6, and then 1 mM isopropyl β-D-1-thiogalactopyranoside (IPTG) was added to induce protein expression at 16 °C for 20 h. The bacteria were harvested by centrifugation and frozen at −80 °C. The bacterial pellets were resuspended with 50 mM Tris-HCl lysate buffers (pH 7.4), and the cells were disrupted by six cycles of sonication (30 s sonication, 60 s pause) on ice. Cell debris and insoluble proteins were pelleted by centrifugation at 4 °C, 12,000 × g for 30 min. The soluble fraction was loaded onto a nickel Sepharose affinity column. The column was washed with washing buffer 1 (50 mM Tris-HCl, 50 mM imidazole, pH 7.8) and then eluted with elution buffer 1 (50 mM Tris-HCl, 250 mM imidazole, pH 7.8). The eluate was further purified by Q Sepharose column by washing with washing buffer 2 (50 mM Tris-HCl, 150 mM NaCl, pH 7.8) and eluting with elution buffer 2 (50 mM Tris-HCl, 400 mM NaCl, pH 7.8). All procedures for protein purification were carried out at 4 °C and the purified Halo-rhEGF protein was analyzed by SDS-PAGE and Coomassie blue staining.

Cell culture and cell proliferation assay

NIH 3T3 and HaCaT cell lines were purchased from ProCell Life Science & Technology (Wuhan, China). Both cell lines were cultured in DMEM supplemented with 10% FBS and 1% penicillin/streptomycin at 37 °C in a 5% CO2 incubator. The mitogenic activity of Halo-rhEGF was assessed using NIH 3T3 and HaCaT cells, with recombinant FGF2 produced previously as a positive control (Sun et al., 2019). The cells in the control group (CON) were treated without growth factors. The cells were seeded in 48-well plates at a concentration of 2,000 cells/well. Once the cell density reached 30%, they were treated for 48 h with varying concentrations (0, 0.31, 1.25, 5, 20, and 80 ng/mL) of Halo-rhEGF and FGF2 diluted with DMEM containing 2.5% FBS. Following the treatment, the cultured cells were fixed with 4% (w/v) PFA for 20 min. The nuclei were stained with DAPI for 15 min and the nucleus number was quantified to determine the effect of Halo-rhEGF and FGF2 on cell proliferation.

Coomassie blue staining

The fixed cells were permeabilized with 0.25% Triton X-100 for 15 min and then immersed in Coomassie blue staining buffer (0.1% Coomassie R-250 in 40% ethanol and 10% acetic acid) for 1 h at room temperature. The stained samples were washed with distaining solution (20% ethanol and 10% acetic acid) for 2 h before visualization with microscope.

Western blot analysis

NIH 3T3 and HaCaT were treated with Halo-rhEGF (20 ng/mL) and FGF2 (20 ng/mL) for 1 h to determine the activation of intracellular signaling proteins. The collected protein samples were separated by 10% SDS-PAGE and then transferred onto a polyvinylidene fluoride membrane (Millipore, Merck, Cork, Ireland). The membrane was blocked with 5% (w/v) skimmed milk in TBST (20 mM Tris, 150 mM NaCl and 0.1% Tween in water, pH 7.5) and incubated with primary antibodies against ERK1/2, phos-ERK1/2, c-Jun, phos-c-Jun, Tubulin, Cyclin D1 (CCND1) at 4 °C for 24 h. The membranes were incubated with HRP-conjugated secondary antibodies at room temperature for 1 h and the labelled protein bands were visualized on Amersham Imager 600 (Amersham Biosciences, Piscataway, NJ, USA). The band intensity was analyzed using ImageJ (version 1.8.0) software (Bethesda, MA, USA).

Cell viability assay

Cells were subjected to treatments with Halo-rhEGF (20 ng/mL) and FGF2 (20 ng/mL) for 24 h and then incubated with 2 µM Calcein AM for 45 min in the incubator. The treated cells were washed with DMEM medium for three times before imaging with an epi-fluorescence microscope (Leica DMI3000 B, Leica, Wetzlar, Germany).

Immunofluorescence imaging

HaCaT was starved with DMEM containing 2.5% FBS for 24 h and then treated with and without 20 ng/mL Halo-rhEGF for another 24 h before fixation. The fixed cells were permeabilized and blocked with 5% (w/v) BSA in TBST. The cells were incubated with primary antibodies (1:200 dilution) overnight at 4 °C and incubated with goat Anti-Rabbit antibody labelled with Alexa Fluor 555 (ab150078) for 1 h. The nuclei were stained with DAPI and the fluorescent images were captured with a laser scanning confocal microscope (FV1000; Olympus, Tokyo, Japan).

Protection of HaCaT from UVB irradiation by treatment with Halo-rhEGF

HaCaT cells were incubated with and without 20 ng/mL Halo-rhEGF for 24 h and then exposed to UVB light (50 mJ/cm2, 312 nm light source) for 10 min. Following UVB irradiation, the cells were incubated with 10 μM (2′,7′-Dichlorodihydrofluorescein diacetate) DCFH-DA at 37 °C for 40 min in the dark for determination of reactive oxygen species (ROS) level. The stained cells were washed with PBS twice and DMEM containing Hoechst 33,342 were added to stain nuclei. The fluorescent images were captured with an epi-fluorescent microscope (Leica DMI3000 B).

RNA sequencing and data analysis

HaCaT and HaCaT samples treated with and without 20 ng/mL Halo-rhEGF were collected with RNAiso Plus (9108) and mRNA was extracted with RNeasy Mini Kit (74104, Qiagen, Hilden, Germany). The isolated RNA was quantified by Nanodrop and qualified by Agilent 2100 Bioanalyzer (Agilent Technologies, Palo Alto, CA, USA). Library preparation and RNA sequencing with Illumina NovaSeq 6000 were processed by GENEWIZ company (Suzhou, China). Three biological replicates and one technical replicate were analyzed for each group and the statistical power of this experimental design, calculated in RNASeqPower is 0.998. The sequencing analysis was processed as described (Sun et al., 2023), and the table of differentially expressed genes is shown in the Supplemental Information. The differentially expressed genes were displayed by Volcano plot and GeneExpression-FoldChange plot in MATLAB (The MathWorks, Natick, MA, USA) (Sun, 2023). The MATLAB code for volcano plot and GeneExpression-FoldChange plot was downloaded from GitHub at https://github.com/hscsun/ScatterFoldChanges.git. Gene ontology (GO) analysis and gene set enrichment analysis (GSEA) were processed and the results were displayed with clusterProfiler package in R(3.6) (Wu et al., 2021).

Fluorescence labeling Halo-rhEGF and its interaction with affinity beads

To label Halo-rhEGF with HaloTag ligands, equal volume of ligand (10 μM Halo-TMR dye: G8251or Halo-AlexaFluor 660 dye: G8471, Promega, Beijing, China) and Halo-rhEGF (2 μM) were mixed and incubated at room temperature protected from light for 1 h. The mixture containing TMR-Halo-rhEGF was incubated with Q Sepharose and IDA-Nickel beads (70501-100, Beaver, Suzhou, China) for 10 min and washed with PBS buffer three times. The beads containing TMR-Halo-rhEGF were visualized by an epi-fluorescence microscopy. The mixture containing Alexa660-Halo-rhEGF was incubated with TEV protease for 24 h and separated by SDS-PAGE for 1 h. The fluorescent image of protein gel was captured by Azure C500 (Azure Biosystems, Dublin, CA, USA).

Stimulation of HaCaT with slowly released Halo-rhEGF

Halo-rhEGF (20 μg/mL, 100 μL) was incubated with 100 μL Q Sepharose in distilled water for 15 min to prepare Q-Halo-rhEGF material. Q-Halo-rhEGF (50 μL) was added into a transwell (Corning; Corning, NY, USA) and washed with PBS for three times. Then, the transwell containing Q-Halo-rhEGF was transferred into a 24-well plate for cell culture. HaCaT cells were then treated with PBS (CON group), 100 ng/mL Halo-rhEGF (Halo-rhEGF group) and Q-Halo-rhEGF in transwell (Q-Halo-rhEGF group) and incubated for 5 days without medium replacement. Calcein AM was used for cell viability determination.

Statistics

The data are presented as means with standard deviation. GRAPHPAD PRISM software version 9.0 (GraphPad Software Inc., San Diego, CA, USA) was applied for statistical analyses. An unpaired Student’s t-test or one-way ANOVA was used for variation analysis.

Results

Production of soluble Halo-rhEGF protein

The DNA sequence of rhEGF was modified for protein expression and the modified DNA sequence was synthesized and sequenced (Figs. 1A and 1B). The sequencing result shows that the corresponding polypeptide sequence of the synthesized gene was consistent with the rhEGF polypeptide sequence (Fig. 1B). The synthesized gene was cloned into pET-14b-Halo vector as described (Fig. 1C), and the recombinant pET-14b-Halo-rhEGF vector was transformed into BL21 (DE3) strain for protein expression (Sun et al., 2015). Protein expression result shows that a strong band around 43 kDa was observed after IPTG induction, which suggests that Halo-rhEGF was successfully expression by BL21 strain (Fig. 1D). After cell disruption by sonication, the soluble fraction was purified by Nickel Sepharose affinity column and Q Sepharose column. It shows that Halo-rhEGF can be easily purified by binding to those both affinity columns (Fig. 1D). The purified Halo-rhEGF was added into HaCaT cell medium to stimulate intracellular signaling, and it shows that both Halo-rhEGF and commercial rhEGF can significantly promote ERK1/2 phosphorylation (Fig. 1E), suggesting that the produced Halo-rhEGF protein possesses similar biological activity to the commercial rhEGF.

Figure 1 Construction of Halo-rhEGF expression vector and production of Halo-rhEGF.

(A) DNA sequence and protein sequence of rhEGF with a hexahistidine tag. (B) DNA sequencing of synthesized human Egf DNA fragment. (C) Scheme of construction of Halo-rhEGF expression vector. (D) Expression and purification of Halo-rhEGF (M: marker, line 1: cell lysate before IPTG induction, line 2: cell lysate after IPTG induction, line 3: pellet, line 4: supernatant, line 5: flow-through, line 6: wash, lines 7–8: elution with 50 mM imidazole, lines 9–10: elution with 250 mM imidazole, lines 11–12: elution with phosphate buffer containing 0.3 M NaCl). (E) Effect of Halo-rhEGF and rhEGF on phosphorylation of ERK1/2. Three biological replicates were performed for Western blot assay (***p < 0.001).

Effect of Halo-rhEGF on HaCaT and NIH 3T3 cell growth

The effect of Halo-rhEGF on HaCaT proliferation was measured by analyzing nucleus number, and the results indicated that 5 ng/mL Halo-rhEGF significantly promoted cell proliferation, and the best effect was achieved at 20 ng/mL (Fig. 2A). Since it was reported that FGF2 has effects on HaCaT, it was used as a positive control (Hahm et al., 2021). The result shows that Halo-rhEGF has a significantly higher ability to promote the proliferation of HaCaT than FGF2 (Fig. 2A). To ensure that the activity of Halo-rhEGF was not restricted to HaCaT cells, the effect on NIH 3T3 proliferation was measured. The results indicated that Halo-rhEGF could also promote the proliferation of NIH 3T3, but FGF2 showed better efficacy than Halo-rhEGF (Fig. 2B). The Coomassie blue staining results indicate that both Halo-rhEGF and FGF2 can promote cell proliferation of HaCaT, and the effect of Halo-rhEGF was more significant (Fig. 2C). Moreover, FGF2 could suppress the adhesion function of NIH 3T3 fibroblasts, which was also found in cardiac fibroblasts and pulmonary fibroblasts (Sun et al., 2023; Tian et al., 2023), but Halo-rhEGF did not change the cell morphology (Fig. 2C). The nucleus staining results show that both Halo-rhEGF and FGF2 could promote the division of the nucleus of HaCaT and NIH 3T3 (Fig. 2D), which is consistent with the findings in Figs. 2A and 2B.

Figure 2 Effect of Halo-rhEGF and FGF2 on HaCaT and NIH 3T3 cell growth.

(A) The effect of Halo-rhEGF and FGF2 on cell proliferation of HaCaT. (B) The effect of Halo-rhEGF and FGF2 on cell proliferation of NIH 3T3. (C) The effect of Halo-rhEGF and FGF2 on cell morphology. (D) The effect of Halo-rhEGF and FGF2 on the division of the nucleus . Three biological replicates were performed for nucleus number analysis. Scale bar: 200 μm.

The activation of intracellular signaling was analyzed to determine the stimulation effect of Halo-rhEGF. The results show that Halo-rhEGF can promote the phosphorylation level of ERK1/2 in both NIH 3T3 and HaCaT, which can also be suppressed by U0126 (MEK1/2 inhibitor). Similar to the cell proliferation results, the effect of Halo-rhEGF on ERK1/2 phosphorylation was weaker than FGF2 in NIH 3T3, while the effect was stronger than FGF2 in HaCaT (Figs. 3A and 3B). The results also show that Halo-rhEGF could also stimulate phosphorylation of c-Jun, which was inhibited by SP600125 JNK inhibitor (Figs. 3A and 3B). The cell viability was analyzed with calcein-AM probe. It shows that both Halo-rhEGF and FGF2 could increase the viability of HaCaT and NIH 3T3, but only a part of the effect was inhibited by U0126 (Fig. 3C). Moreover, the regulatory effect of Halo-rhEGF on HaCaT was stronger, whereas FGF2 was more potent on NIH 3T3 (Fig. 3C). In the previous studies, it was found FGF2 can regulate the proliferation and activation of primary fibroblasts isolated from heart and lung (Sun et al., 2022, 2023; Tian et al., 2023). Thus, EGF is a better agent to promote the proliferation and repair of skin epidermal cells, while FGF2 is a better agent to regulate the proliferation of fibroblasts and the combination of these two drugs is a better choice for the repair of skin tissue damage.

Figure 3 Effect of Halo-rhEGF and FGF2 on intracellular signaling activation and cell viability.

(A and B) Western blot analysis of phosphorylation of ERK1/2 and c-Jun. (C) Effect of Halo-rhEGF and FGF2 on cell viability of HaCaT and NIH 3T3. Three biological replicates were performed for Western blot assay and cell viability assay. The p values of the comparison between control group and Halo-rhEGF group are less than 0.05, and the p values of the comparison between Halo-rhEGF group and Halo-rhEGF+inhibitor group are also less than 0.05. U is a MEK1/2 inhibitor (U0126), and SP is a JNK inhibitor (SP600125). Scale bar: 200 μm.

RNA sequencing revealed the biological functions of Halo-rhEGF on HaCaT

RNA sequencing was applied to determine the biological functions of Halo-rhEGF on HaCaT. The result shows that 684 genes were up-regulated and 579 genes were down-regulated when the log2(fold change) parameters were set to ±0.5 (Fig. 4A). GSEA result reveals that multiple biological functions were promoted, which include DNA replication and metabolic, ribosome biogenesis, mRNA processing and protein folding (Fig. 4B). GO analysis also indicates that the differentially expressed genes are enriched to the functions on cell growth, epithelial cell proliferation, ribosome biogenesis and so on (Fig. 4C). The regulated genes enriched in ribosome biogenesis and epithelial cell proliferation are displayed and most genes in these two functions were up-regulated by Halo-rhEGF (Fig. 4D). Taken together, Halo-rhEGF can promote the proliferation and viability of HaCaT by regulating gene transcription.

Figure 4 Regulation of gene expression and biological function of HaCaT by Halo-rhEGF.

(A) Volcano plot of differentially expressed genes regulated by Halo-rhEGF. (B and C) The trends in the regulation of biological functions enriched by GSEA (B) and GO analysis (C). (D) The differentially expressed genes on ribosome biogenesis and epithelial cell proliferation. Three biological replicates were performed for RNA sequencing analysis.

The sequencing result was validated by analyzing the expression levels of genes and proteins in the list of differentially expressed genes (Figs. 5A and 5B). It shows that the gene expression analyzed by qPCR is consistent with the RNA sequencing results, suggesting that the RNA sequencing results is reliable (Fig. 5A). The expression levels of multiple proteins related to DNA replication and cell proliferation, e.g. c-Jun, PCNA, HMGA1 and MCM2, were promoted by Halo-rhEGF (Figs. 5B and 5C). The cell immunofluorescence results also indicate the protein levels of c-Jun and CCND1 were increased by Halo-rhEGF (Figs. 5D and 5E).

Figure 5 Effect of Halo-rhEGF on the expression of proteins related to cell proliferation.

(A) Validation of the differentially expressed genes by q-PCR with eight replicates. (B and C) Western blot analysis of the expression of proteins related to cell proliferation. (D and E) Cell immunofluorescence detection of the protein levels of c-Jun and CCND1. Three biological replicates were performed for Western blot assay. The p values of the comparison between control group and Halo-rhEGF group are all less than 0.05. Scale bar: 200 μm.

Halo-rhEGF attenuates UVB-induced damage in HaCaT

UVB irradiation was applied to treat HaCaT to induce its damage, and Halo-rhEGF was added for 24 h before irradiation to protect HaCaT from the damage. After UVB irradiation, the cell damage was quantified by determining ROS level with DCFH-DA probe as previously described (Hegedűs et al., 2021; Li et al., 2021; Long et al., 2023). The result shows that DCF fluorescence was increased after UVB irradiation and Halo-rhEGF can suppress this increasement, suggesting Halo-rhEGF can inhibit the generation of intracellular ROS (Fig. 6A). The cell viability assay shows that UVB irradiation can decrease cell viability, while Halo-rhEGF inhibited the decrease of cell viability (Fig. 6B). Therefore, Halo-rhEGF is a potential biopharmaceutical for promoting the viability and regenerative capacity of keratinized epithelial cells.

Figure 6 Halo-rhEGF attenuates UVB-induced damage in HaCaT.

(A) The effect of Halo-rhEGF on suppression of ROS induced by UVB irradiation. (B) The effect of Halo-rhEGF on protection of cell viability. (C) Mechanism of Halo-rhEGF on the regulation of cell growth and survival, created using the shapes and icons included in Microsoft PowerPoint (version 16.86). Three biological replicates were performed for cell viability assay and ROS assay (*p < 0.05, **p < 0.01, ***p < 0.001). Scale bar: 200 μm.

HaloTag for fluorescence labeling and binding to biomaterials

HaloTag is a gene-engineered protein for chloroalkane fluorescent ligand labeling, which can used to visualize the protein of interest (Zhang et al., 2006; Los et al., 2008; Frei et al., 2022). The SDS-PAGE result shows that Halo-rhEGF can be labelled by Halo-AlexaFluor 660 dye and TEV protease can cleave Halo-rhEGF into HaloTag and rhEGF (Fig. 7A). Meanwhile, the fluorescence imaging results show that TMR-Halo-rhEGF can bind to both Q Sepharose and IDA-Nickel beads, which indicate that Q Sepharose and IDA-Nickel beads are desired materials for Halo-rhEGF protein purification (Fig. 7B).

Figure 7 Application of HaloTag for fluorescence labeling and biomaterial binding.

(A) Labelling of Halo-rhEGF with AlexaFluor 660 ligand and cleavage of Halo-rhEGF with TEV protease. (B) Labelling of Halo-rhEGF with TMR ligand and binding of TMR-Halo-rhEGF to Q Sepharose and IDA-Nickel beads. (C and D) Effect of sustained-release of Halo-rhEGF on cell morphology and viability. Three biological replicates were performed for cell viability assay (*p < 0.05, **p < 0.01, ***p < 0.001). Scale bar: 200 μm.

Since HaloTag can bind to Q Sepharose by ion interaction, Q Sepharose can also be used as a biomaterial for controlled release of Halo-rhEGF (Q-Halo-rhEGF). The results show that both Halo-rhEGF and Q-Halo-rhEGF can promote cell proliferation and viability of HaCaT, but Q-Halo-rhEGF can also be able to maintain the junctions between cells and can enhance cell viability more strongly (Figs. 7C and 7D). Therefore, HaloTag can also be used as a protein tag for sustained release of rhEGF protein, which is appropriate for the development of dressings containing rhEGF drugs in the future.

Discussion

The HaloTag protein was initially used for fluorescence labeling and real-time tracking of interested proteins due to its convenient operation and compatibility with a variety of different fluorescent probes (Los et al., 2008; Frei et al., 2022). In our study, we utilized HaloTag for the expression of growth factors and found that the co-expression of HaloTag protein with FGF or EGF could enhance their solubility (Sun et al., 2015). Additionally, HaloTag protein carries a certain amount of negative charge on its surface, allowing it to bind to strong anion-exchange resins, providing a method for further purification of the protein (Sun et al., 2015). Furthermore, Halo-rhEGF protein designed in this study contains a C-terminal hexahistidine tag, and the binding of Halo-rhEGF to nickel ion resin confirmed the integrity of Halo-rhEGF protein sequence. Therefore, the Halo-rhEGF prepared in this study has good solubility and integrity, which ensures the biological activity of the rhEGF protein drug.

The previous studies have shown that EGF could be used to promote the growth and proliferation of epithelial cells and has been used in various skin care products (Hori et al., 2007; Yoo et al., 2014; Cheng et al., 2020). In this study, the cellular experimental results showed that Halo-rhEGF can significantly promote the cell viability and proliferation of epithelial cells and fibroblasts by regulating ERK1/2 signaling and c-Jun signaling, and its effect is similar to the commercial recombinant EGF protein. Moreover, Halo-rhEGF has a stronger regulatory effect on epithelial cells than FGF2, but FGF2 has a better regulatory effect on fibroblasts than Halo-rhEGF. The findings suggest that Halo-rhEGF is effective in enhancing the activity and proliferation of epithelial cells, offering a potential application in regenerative medicine and tissue engineering. Its superior activity compared to FGF2 in regulating epithelial cells indicates that Halo-rhEGF could be a valuable tool for promoting epithelial cell function and wound healing processes. However, the preference of FGF2 for fibroblast regulation suggests that the choice of growth factor should be tailored to the specific cellular context and the desired outcome in fibroblast-rich environments.

Moreover, this study also showed that Halo-rhEGF could protect HaCaT cells from the damage by UVB radiation, providing evidence of its protective effect on epithelial cells and offering a method for skin care. But the findings of this study could be further validated through preclinical and clinical trials to determine the precise therapeutic applications of Halo-rhEGF in skin care products and other dermatological conditions. Such research could also explore the potential synergies between Halo-rhEGF and other skin care ingredients or therapeutic agents to enhance the efficacy and safety of skin care products.

Conclusions

In this study, we cloned human Egf gene into pET-14b-HaloTag plasmid vector to produce Halo-rhEGF. Most of the expressed Halo-rhEGF was in the soluble fraction, and successfully purified by Nickel affinity column and anion exchange column. Further analysis indicates that Halo-rhEGF can active the intracellular signaling to promote cell proliferation and increase cell viability. Meanwhile, Halo-rhEGF can regulate the biological activity of epithelial cells more effectively, and is more suitable as a protein drug for the protection of superficial epithelial cells. This study provides a new scheme for the preparation, activity detection and sustained release of rhEGF protein drugs in clinical practice.

Supplemental Information

Supplemental Information 1 The differentially expressed genes (log2 fold change parameters: ±0.5).

Supplemental Information 2 Uncropped Gel Images.

Supplemental Information 3 Raw numerical data for statistics.

Additional Information and Declarations

Competing Interests

Author Contributions

Data Availability

The authors declare that they have no competing interests. Yezhuo Liu is employed by Shanghai Hongdu New Material Science and Technology and Favorsun Medical Group. Changye Sun is the part-time technical director of Shanghai Hongdu New Material Science.

Mengru Bai performed the experiments, analyzed the data, prepared figures and/or tables, authored or reviewed drafts of the article, and approved the final draft.

Yezhuo Liu conceived and designed the experiments, analyzed the data, authored or reviewed drafts of the article, and approved the final draft.

Hongyin Liu analyzed the data, prepared figures and/or tables, authored or reviewed drafts of the article, and approved the final draft.

Yangyang Jia analyzed the data, authored or reviewed drafts of the article, and approved the final draft.

Xiangqin Tian analyzed the data, authored or reviewed drafts of the article, and approved the final draft.

Changye Sun conceived and designed the experiments, performed the experiments, analyzed the data, prepared figures and/or tables, authored or reviewed drafts of the article, and approved the final draft.

The following information was supplied regarding data availability:

The raw sequence files for mRNA sequencing are available at NCBI Sequence Read Archive: PRJNA1068279.

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
