# Peer review of "Production of recombinant human epidermal growth factor fused with HaloTag protein and characterisation of its biological functions"

_PeerJ, doi:10.7717/peerj.17806_

## Round 0.1 · original submission · Major Revisions

Dear authors, many thanks for you submission and relevant work. Presently, it still requires significant revisions and clarifications. Please, refer to the reviewers comments for further details. Do not forget to be very clear with your methodology (including reference numbers of kits and materials where relevant.).

Reviewer 1 ·

Basic reporting

no comment

Experimental design

no comment

Validity of the findings

no comment

Additional comments

1. Please provide the statistics in the form of p-values for all the data provided.
2. Figure 2. Please provide a better description for the figure. For eg. Fig 2 A is for the HACAT cells and figure 2 for NIH-3T3 cells.
3. Figure 2C is it for cell morphology or proliferation?
4. Line 216 “Halo-rhEGF showed better efficacy than FGF2 (Fig. 2B).”. In the graphical presentation it is shown that FGF2 has better efficacy, if so, please make the appropriate correction.
5. Line 219 please provide the inhibitor U0126 detail.
6. Figure 3 please provide the inhibitor information in the figure legend.
7. For the immunofluorescence images are the same display settings used for the control and recombinant protein.
8. Line 116 are the different concentrations of recombinant protein made in water.
9. Figure 3D what is the criteria for measuring the fluorescence intensity. Is some specific area on the image chosen or the whole image.
10. Figure 6 please check the figure legend and figures details. Figure 6A is for cell proliferation and 6B for ROS.
11. Line 208 please provide details for figure 2C and D in the text.
12. For western blot and other immunofluorescence assay what doses were used for stimulating cells with recombinant proteins.
13. Is any cytotoxicity study done on the halo-EGF. In this paper 80 ng/mL max dose is used it does not show drastic proliferation beyond 20 ng/mL. Is any other higher dose is also tested that is cytotoxic for the cells.

Reviewer 2 ·

Basic reporting

In this manuscript, Bai et al. developed a Halo-tag-based approach for purifying epidermal growth factor (EGF). The authors also tested the role of EGF in gene transcription and cell proliferation. Overall, the research was well-conducted; however, the novelty of this work does not meet the standards of PeerJ, and the current version of this manuscript is not recommended for publication. I will provide more comments below:
1. The English language should be improved to ensure clarity for an international audience. Examples of areas needing improvement include lines 55, 222, and 267, where the current phrasing hinders understanding. I recommend having a native speaker review your manuscript or engaging a professional editing service.
2. In the abstract, the sentence 'However, EGF protein expressed in prokaryotic cells often occurs in inclusion bodies, and the auxiliary expression of soluble tag protein is an efficient method to prepare EGF protein' is unclear. Please rephrase it to make it easier to understand.
3. The sentence 'These tag proteins have a limited effect on improving the solubility of expressed EGF in E. coli and are not of great value for purification and later studies of recombinant EGF protein (Kim et al., 2021; Ferreira et al., 2022)' seems to be improperly cited, as I do not find any evidence that supports the assertion that SUMO and MBP tags are unable to increase solubility, or that their effects are 'limited'. Please provide direct evidence from the reference. Furthermore, I recommend providing experimental data that directly compares the Halo-tag with other soluble tags to support that the Halo-tag is a better option.
4. The abbreviation 'FGF2' is missing.
5. In Fig 4B, what does the 'CON' group represent?

Experimental design

no comment

Validity of the findings

no comment

Reviewer 3 ·

Basic reporting

The paper proposes the utilization of a new protein fusion for bacterial expression of a functional form of the EGF domain of the human EGF growth factor. The fusion protein used is a modified form of a bacterial haloalkane dehalogenase. The fusion protein is named Halo-rhEGF throughout the paper. The results presented demonstrate an efficient purification of the recombinant Halo-rhEGF and its use in experiments to stimulate cell proliferation. The authors proceed to show that Halo-rhEGF is functional by analyzing the activation of some targets of the EGF signaling pathway, including phosphorylation of the kinases ERK1/2 and of c-Jun. Additionally, the authors demonstrate the utilization of Halo-rhEGF coupled to a chromophore to both label and induce cells expressing EGF receptors.
The methods and the data presented are in general consistent with appropriate controls. However, the investigation of Halo-rhEGF function was complemented by gene expression analysis of cells induced with this fusion protein by using large scale RNA sequencing. This became a weak point for the paper because the authors describe in the methods (see also Figure 4A) that the fold change used to classify differentially expressed was 0.5. Usually, a minimum fold change of 2 is required to classify differentially expressed genes since all changes in gene expression below a fold change of 2 is considered not relevant. This point must be reviewed and depending on the new parameters applied, the whole gene expression analysis may have to be reinterpreted.
Concerning literature citations and introductory information, the authors should make it clear in the beginning of the second paragraph of the introduction that the EGF domain is synthesized as precursor protein that undergoes processing to produce the functional EGF domain. A reference or references for this information should be cited.
More importantly, the second part of the second paragraph of the introduction is not correct [(There are multiple strategies to co-express EGF with soluble proteins, such as thioredoxin, SUMO, maltose binding protein, Fh8 and glutathione S-transferase (Kim et al., 2021; Ferreira et al., 2022). These tag proteins have limited effect on improving the solubility of expressed EGF in E. coli and are not of great value for purification and later studies of recombinant EGF protein (Kim et al., 2021; Ferreira et al., 2022)]. This description is not correct. All these fusion proteins make an important contribution for improving EGF solubility and have been a useful tool to easily produce functional EGF domains in E. coli in the same way as does the Halo-rhEGF fusion. The authors must review their statements regarding the papers cited in this paragraph. Admitting that they are useful is not going to change the conclusion that the Halo-rhEGF fusion is also useful.
Concerning the use of the English language, the text is mostly well written understandable. However, there are quite a few parts that need to be improved and some words are not always used with their proper meaning. Below are some examples but I recommend the whole text to be reviewed to be more precise:
- Line 44: the “The” at the start of the “The previous studies…” sentence is not necessary.
- Line 47: “Prokaryotic expression system is a high yield and cost-effective protein expression system” could be changed to “Prokaryotic protein expression systems can reach high yield and be cost-effective”.
- Lines 62/62: “Moreover, there is (not there are) a certain amount of 63 negatively charged residues…”.
- Lines 69/70: “In addition, HaloTag 70 proteins can bind to cation-rich biomaterials to regulate the sustained release of EGF.” – the expression “to regulate the sustained…” seems not appropriate in this case.
- Lines 71/72: “Thus, HaloTag is more than just a tag protein, which can also regulate the soluble expression and sustained release of co-expressed proteins.” The sentence has no correct meaning to me. The fusion with the Halo protein does not regulate soluble expression. We don’t know the mechanism, but possibly fusion proteins functions as a chaperone during protein folding so that its folding helps the EGF domain also to acquire a native folding. The part “…sustained release of co-expressed proteins” also does not seem to have a correct meaning.
- Line 92. It should be “E. coli.”
- Line 103. It should be “nickel” Sepharose…, not “Nickle” this appear always when nickel is meant.
- Lines 103/104: “The column was washed with washing buffer…”. Repeating wash and washing is not necessary.
- Line 133: HaloEGF should be Halo-rhEGF.
- Lines 144/145: “The fixed cells were permeabilized blocked with 5%”. An “and” is missing between permeabilized and blocked.
- Lines 145/146: “…primary antibodies (1:200 dilution) for overnight…”. For is nor required.
- Lines 164-166: The citation should be placed just after the expression “as described” not at the end of th sentence: The sequencing analysis was processed as described (Sun et al., 2023) and the table of differentially expressed genes is shown in the Supplemental Information
-Lines 174/175: “…at room 175 temperature in dark for 1 hour”. Try: “at room temperature protected from light for 1 hour.
- The first paragraph of the results, lines 194 to 202 need to be rewritten. As for figure 1, parts A to C, the sequence confirmation and the plasmid map don’t need to go into the result section. They can be provided as supplementary information.
- Lines 217 and 223. The past perfect verb “was detect” seems wrong here. It looks more like “was analyzed” in both cases.
- Line 256: when the authors say: “…the recombinant Halo-rhEGF is a potential protein drug...”, it is not clear what is meant by “potential protein drug“, could it be “a potential biopharmaceutical…”?

Experimental design

The methods section is clearly written and easy to understand. There is however major problem with a parameter used to classify the differentially expressed genes in the RNA sequencing analysis. “The fold change parameter was set to +/-0.5. This is not sufficient to distinguish genes that are differentially expressed form those whose expression does not change. Usually, a minimum fold change of 2 is required to classify differentially expressed genes. The authors should carefully review this point because it is the most critical problem of the manuscript. Once this is clarified, the number of differentially expressed genes can change a lot affecting the conclusions drawn from the subsequent analyses with this data.
In addition, an accession number must be given for the sequence used to extract the Asn971-Arg1023 segment used to design the Halo-rhEGF clone.

Validity of the findings

The paper proposes the utilization of a new protein fusion for bacterial expression of a functional form of the EGF domain of the human EGF growth factor. The results show that the EGF domain of the new fusion protein is functionally active and propose additional applications for the Halo-rhEGF fusion protein which in principle cannot be achieved with the previous protein expressed in E. coli in fusion with the EGF domain of the human EGF growth factor. The paper adds up to the variety of options to produce a growth factor of complex tertiary structure in E. coli, which is the most accessible expression system to produce recombinant proteins.

---

## Round 0.2 · Minor Revisions

Dear authors, thank you for your re-submission and work. Some minor aspects need to be revised before i can send the approval. Please check the reviewers' comments, as well.

Additionally (I recognize I probably should have flagged this before), you need to make sure that your methods can be reproduced by other researchers, if they wish so; as such, please, indicate the ref/catalog numbers of key materials such as Kits used, enzymes, antibodies, bacterial strains, etc , and instruments brand+models (if it applies; I think in most cases you have done so correctly already).

Reviewer 1 ·

Basic reporting

The authors have answered my questions, and the paper has significantly improved.

Experimental design

no comment

Validity of the findings

no comment

Additional comments

no c omment

Reviewer 2 ·

Basic reporting

The authors addressed all my concerns, and the current version of this manuscript is ready for publication.

Experimental design

no comment

Validity of the findings

no comment

Reviewer 3 ·

Basic reporting

This review referes to the revised manuscript.
The authors have made all modifications requested and in may view the revised version fo the manuscript can be acceted for publication.
I just have a couple of minor points:
Minor point to correct in the text:
Abstract:
“The intracellular signaling proteins, ERK1/2 and c-Jun, were phosphorylated by Halo-rhEGF….”
- Since Halo-rhEGF doesn’t have intrinsic kinase activity, the correct sentence should be like “Phosphorylation of the intracellular signaling proteins, ERK1/2 and c-Jun, was simulated by treatment with Halo-rhEGF….”
Introduction:
“The precursor of human EGF is constructed of 1207 amino acids containing multiple EGF-like peptides and a 6 kDa EGF polypeptide…”
The word “constructed” is not appropriate, try “The precursor of human EGF comprises 1207 amino acids containing multiple EGF-like peptides and a 6 kDa EGF polypeptide…”

Experimental design

Nothing additional to declare.

Validity of the findings

Nothing additional to declare.

---

## Round 0.3 · Minor Revisions

Dear authors,

i think you missed the part where it was asked for you to add additional information to the methods, namely ref numbers of kits/antibodies, other key proteins etc...

Example when you look for NcolI in thermofisher's catalog: https://www.thermofisher.com/search/results?query=NcoI%20&focusarea=Search%20All .
This is a minor revision but important to assure reproducibility and/or replicability capacities.

---

## Round 0.4 · accepted · Accept

Dear authors,

Thank you for you submission to PeerJ and your hard-work. Your manuscript is now accepted for publication.